# Developments and Performance of Artificial Intelligence Models Designed for Application in Endodontics: A Systematic Review

**DOI:** 10.3390/diagnostics13030414

**Published:** 2023-01-23

**Authors:** Sanjeev B. Khanagar, Abdulmohsen Alfadley, Khalid Alfouzan, Mohammed Awawdeh, Ali Alaqla, Ahmed Jamleh

**Affiliations:** 1Preventive Dental Science Department, College of Dentistry, King Saud Bin Abdulaziz University for Health Sciences, Riyadh 11426, Saudi Arabia; 2King Abdullah International Medical Research Centre, Ministry of National Guard Health Affairs, Riyadh 11481, Saudi Arabia; 3Restorative and Prosthetic Dental Sciences Department, College of Dentistry, King Saud Bin Abdulaziz University for Health Sciences, Riyadh 11426, Saudi Arabia

**Keywords:** machine learning, deep learning, artificial neural network, conventional neural network, root canal treatment, apical lesions, diagnosis, detection, prediction

## Abstract

Technological advancements in health sciences have led to enormous developments in artificial intelligence (AI) models designed for application in health sectors. This article aimed at reporting on the application and performances of AI models that have been designed for application in endodontics. Renowned online databases, primarily PubMed, Scopus, Web of Science, Embase, and Cochrane and secondarily Google Scholar and the Saudi Digital Library, were accessed for articles relevant to the research question that were published from 1 January 2000 to 30 November 2022. In the last 5 years, there has been a significant increase in the number of articles reporting on AI models applied for endodontics. AI models have been developed for determining working length, vertical root fractures, root canal failures, root morphology, and thrust force and torque in canal preparation; detecting pulpal diseases; detecting and diagnosing periapical lesions; predicting postoperative pain, curative effect after treatment, and case difficulty; and segmenting pulp cavities. Most of the included studies (*n* = 21) were developed using convolutional neural networks. Among the included studies. datasets that were used were mostly cone-beam computed tomography images, followed by periapical radiographs and panoramic radiographs. Thirty-seven original research articles that fulfilled the eligibility criteria were critically assessed in accordance with QUADAS-2 guidelines, which revealed a low risk of bias in the patient selection domain in most of the studies (risk of bias: 90%; applicability: 70%). The certainty of the evidence was assessed using the GRADE approach. These models can be used as supplementary tools in clinical practice in order to expedite the clinical decision-making process and enhance the treatment modality and clinical operation.

## 1. Introduction

The specialty of endodontics deals with the diseases and conditions that affect the root canal complex and are developed due to untreated or incompletely treated dental carious lesions [1,2]. Diseases related to the pulp and periapical tissues are most commonly managed by nonsurgical root canal treatment (RCT). The basis of endodontic diagnosis and treatment planning relies on an adequate and accurate understanding of the diseases related to the pulp and periapical tissues. Inaccurate diagnosis may result in unanticipated pain, which may have a negative impact on the therapeutic plan and eventually result in unpleasant experiences among patients [3]. Preoperative assessment of the tooth, before initiating RCT, is a very crucial step in determining the success of the endodontic treatment.

Intraoral periapical radiographs, orthopantomograms, and cone-beam computed tomography (CBCT) imaging are the most frequently adopted radiographic techniques for diagnosing diseases related to pulp and periapical areas [2]. Periapical and panoramic radiographs generate two-dimensional (2D) images of the maxillofacial structures, with lesser exposure than CBCT imaging [4,5]. CBCT imaging is widely used among dentists as it enables more radiological analysis. This technology provides three-dimensional images with more precision [6]. The accuracy in detecting periapical lesions is significantly higher with CBCT imaging in comparison to periapical radiography [7,8]. However, considering its high cost and radiation dose, the use of CBCT imaging is restricted to special clinical circumstances. In such cases, the benefits obtained from the imaging should outweigh any potential risks resulting from radiographic exposure associated with this technology.

The ongoing rapid technological advancements have resulted in enormous development in diagnostic models for medical imaging and diagnosis [9]. Advancements in computer-assisted diagnosis have resulted in the development of AI models designed for application in health sectors. AI technology, which is mainly based on mimicking the functioning of the human brain, is a breakthrough in the technological world. Machine learning algorithms were the first AI algorithms developed, the performance of which is dependent on the characteristics and number of datasets used for training. These algorithms are utilized to learn the intrinsic statistical patterns and structures in the data and are later applied for making predictions when applied to unseen data [10]. Deep learning (DL) or convolutional neural networks (CNNs) are developed to mimic the functioning of the human brain; they are designed to solve equations by passing through a series of convolutional filters and are trained on a large number of datasets [11]. These advanced neural networks are applied for processing large and complex images, where they have demonstrated superior achievements in recognizing objects, faces, and activity [12,13]. AI models have been widely applied in medical imaging for systemic diseases such as cardiovascular diseases and respiratory diseases and have displayed exceptional performances that are similar to those of experienced specialists [14,15,16]. Additionally, in dentistry, AI models are designed for diagnosing oral diseases such as dental caries, periodontal diseases, and oral cancer as well as treatment planning for orthognathic surgeries and predicting the treatment outcomes [17,18,19]. These models have demonstrated excellent performances, with a major advantage of this being improved diagnostic efficiency with a reduced image interpretation time [20]. Nagendrababu et al. [21] reported on AI models designed for application in endodontics for performing tasks such as studying root canal anatomy, detecting and diagnosing periapical lesions and root fractures, and determining the working length for planning root canal treatment. The authors concluded that these AI models can aid clinicians with precise diagnosis and treatment planning, ultimately resulting in better treatment outcomes. Umer et al. [22] also reported on AI models designed for application in endodontic diagnosis and treatment planning. The authors concluded that these models demonstrated an accuracy greater than 90% in performing the tasks. However, the authors also stated that the reporting of AI-related research is irregular. Hence this systematic review aimed to report on the application and performances of AI models designed for application in endodontics.

## 2. Materials and Methods

Ethical clearance was obtained from King Abdullah International Medical Research Center (Institutional Review Board Approval No. 2439-22, 6 November 2002) before the literature search process was initiated for this systematic review.

The updated Preferred Reporting Items for Systematic Reviews and Meta-analyses (PRISMA) guidelines were considered for preparing this systematic review [23]. A search of the literature was conducted systematically in various renowned electronic databases, primarily Scopus, Web of Science, Embase, PubMed, and Cochrane and secondarily the Saudi Digital Library and Google Scholar, for studies relevant to the research topic that were published from 1 January 2000 to 30 November 2022.

### 2.1. Search Strategy

The article search was performed based on the research question, which was developed in accordance with the PICO elements (P: problem/patient/population, I: intervention/indicator, C: comparison, and O: outcome).

Research Question: What are the developments, applications, and performances of AI models in endodontics?

Population: Patients who underwent investigation for endodontic diagnosis (dental radiographs such as intraoral periapical, bitewing, occlusal, and panoramic radiographs; cephalograms; cone-beam computed tomography (CBCT); digital photographs; 3D CBCT images).

Intervention: Artificial intelligence applications that were designed for the detection, diagnosis, and prediction of endodontic lesions.

Comparison: Various reference standards, testing models, expert/specialist interpretations

Outcome: Predictable or measurable outcomes such as accuracy, specificity, sensitivity, receiver operating characteristic curve (ROC), area under the curve (AUC), area under the receiver operating characteristic (AUROC), intersection over union (IOU), intraclass correlation coefficient (ICC), statistical significance, F1 scores, volumetric Dice similarity coefficient (vDSC), surface Dice similarity coefficient (sDSC), positive predictive value (PPV), negative predictive value (NPV), and Dice coefficient.

Medical Subject Headings (MeSH) included artificial intelligence, automatic learning, supervised learning, unsupervised learning, deep learning, machine learning, neural networks, convolutional neural network, computer-assisted diagnosis, endodontic dentistry, root canal treatment, apical lesions, periapical lesions, periapical pathology, periapical diseases, deep caries detection, tooth segmentation, pulp cavity segmentation, root segmentation, root morphology, canal shape, cracked tooth, tooth fractures, root fractures, accuracy, prediction, and diagnosis. Boolean operators such as and/or were also used in the advanced stage of the search for combining these MeSH terms, with predetermined publication time range and language as filters (Table 1).

Simultaneously, a manual search for articles was also performed by cross-referencing and screening the bibliography list of the selected articles.

### 2.2. Study Selection

The article selection process was carried out in two phases. In the first stage, the articles that were related to the research question were selected based on their title and abstract. In this phase, two experienced authors (S.B.K. and A.O.J.) simultaneously carried out the search process and 264 articles were selected. After screening, 124 articles were eliminated due to duplication, and the rest of the articles (140 articles) were assessed for meeting the eligibility criteria

### 2.3. Eligibility Criteria

The inclusion criteria were the following: (a) original research articles with a clear statement on AI applications designed for endodontics; (b) articles published between 1 January 2000 and 30 November 2022 in a scholarly peer-reviewed journal; (c) articles with a clear mention of a type of study modality used for developing, training, validating, and testing an AI model; (d) articles with a clear mention of quantifiable outcome measures for assessing the performance of the AI model; (e) AI models applied for determining working length, vertical root fractures, or root morphology or for detecting and diagnosing pulpal diseases, periapical lesions, predicting prognosis, postoperative pain, or case difficulties. The study design was not limited and hence did not affect the articles’ inclusion.

The determined exclusion criteria were as follows: (a) non-full-text articles with only abstracts; (b) non-peer-reviewed publications (such as conference papers and unpublished thesis projects); (c) review articles, letters to editors, and commentaries.

### 2.4. Data Extraction

After the preliminary evaluation of the selected papers based on the title and abstract and the elimination of the duplicates, the authors further analyzed the full text of these articles and assessed their eligibility, following which the total number of articles included in this systematic review decreased to 38. Following that, in the second phase, the identifiers of the journal and author details were removed, and the articles were distributed for critical evaluation by two independent authors who did not contribute to the initial search (M.A. and K.A.). The data from these included articles were further extracted and entered into a Microsoft Excel sheet. This data comprised details of the authors; year of publication; objective of the study; type of algorithm used for developing the AI model; data used for training, validating, and testing the model; results; conclusions; and suggestions.

The quality assessment of the articles was conducted utilizing the Quality Assessment and Diagnostic Accuracy Tool (QUADAS-2) guidelines [24]. This tool was developed to assess the quality of studies that have reported on diagnostic tools. The assessment is based on four domains (patient selection, index test, reference standard, and flow and timing), each of which is evaluated for risk of bias and applicability. The inter-rater reliability between the two authors was assessed on a sample of articles, where Cohen’s kappa showed 86% agreement. The authors had a disagreement regarding the inclusion of one article since the quantifiable outcome measures of performance were not clearly mentioned. This was further resolved through a third opinion obtained (A.F.), after which the article was excluded. Thirty-seven articles finally underwent qualitative synthesis (Figure 1).

## 3. Results

The qualitative data synthesis was performed on the 37 articles [25,26,27,28,29,30,31,32,33,34,35,36,37,38,39,40,41,42,43,44,45,46,47,48,49,50,51,52,53,54,55,56,57,58,59,60,61] that fulfilled the inclusion criteria. The research trend shows there has been a gradual increase in the number of research publications that have reported on the application of AI in endodontics.

### 3.1. Qualitative Synthesis of the Included Studies

In endodontics, AI models have been applied for determining working length (*n* = 3) [25,26,38], determining VRFs (*n* = 6) [27,29,30,32,41,60], detecting pulpal diseases (*n* = 2) [28,42], detecting and diagnosing periapical lesions (*n* = 13) [31,35,36,37,40,43,44,49,50,52,54,55,61], determining root morphology (*n* = 4) [33,39,45,56], predicting postoperative pain (*n* = 1) [48], determining root canal failures (*n* = 4) [51,57,58], predicting case difficulty (*n* = 1) [34], determining thrust force and torque in canal preparation(*n* = 1) [46], segmenting pulp cavities (*n* = 1) [47], and predicting curative effect after treatment (*n* = 1) [53,59].

The data from these included articles were extracted. However, due to the heterogeneity in the data extracted from these articles, performing a meta-analysis was not possible. The heterogeneity was mainly with respect to the different types of data samples applied for assessing the performance of AI models. Hence, in this systematic review, only the descriptive data of the included studies are presented (Table 2).

### 3.2. Study Characteristics

The study characteristics extracted from the included studies included details of the authors; year of publication; objective of the study; type of algorithm used for developing the AI model; data used for training, validating, and testing the model; results; conclusions; and suggestions.

### 3.3. Outcome Measures

The outcome was measured in terms of task performance efficiency. The outcome measures were reported in terms of accuracy, sensitivity, specificity, receiver operating characteristic curve (ROC), area under the curve (AUC), area under the receiver operating characteristic curve (AUROC), intraclass correlation coefficient (ICC), intersection over union (IOU), precision–recall curve (PRC), statistical significance, F1 scores, volumetric Dice similarity coefficient (vDSC), surface Dice similarity coefficient (sDSC), positive predictive value (PPV), negative predictive value (NPV), mean decreased Gini (MDG) coefficient, mean decreased accuracy (MDA) coefficient, and Dice coefficient.

### 3.4. Risk of Bias Assessment and Applicability Concerns

Assessment of the quality of the included studies through the risk of bias is essential in order to understand and report the selection of the samples, reference standards, and methods applied for validating and testing the models.

A low risk of bias was observed in the patient selection domain in most of the studies (risk of bias: 90%; applicability: 70%). However, cadaver samples (Saghiri et al. [25]), extracted teeth (Saghiri et al. [26], Kositbowornchai et al. [27], Johari et al. [29], Qiao et al. [38]), and bone samples (Guo et al. [46]) had been utilized in six studies. Therefore, the patient selection domain of the applicability arm of the tool for these above-mentioned studies was reported to have a high risk of bias. Index tests were regarded as low risk in both the arms of QUADAS-2 since all the studies had made use of a highly standardized system of AI for training purposes. There was no clear mention of the reference standard for interpreting index test results in four of the included studies, which raised concerns regarding bias related to patient selection, reference standard, flow, and timing of these studies in both arms. Overall, there was a low risk of bias in both arms, considering all the categories across the included studies. Details about the risk of bias assessment using QUADAS-2 are mentioned in the Appendix A and Figure 2.

### 3.5. Assessment of Strength of Evidence

The certainty of the selected studies in the systematic review was assessed using the Grading of Recommendations Assessment Development and Evaluation (GRADE) approach [62]. Risk of bias, inconsistency, indirectness, imprecision, and publication bias are major domains under which the certainty of the evidence is rated and categorized as very low, low, moderate, or high. Overall, the studies included in this systematic review showed moderate evidence (Table 3).

## 4. Discussion

Technological advancements in health sciences have led to enormous developments in the AI models that have been designed for application in health sectors. In recent developments, CNN-based AI models have demonstrated excellent efficiency in diagnosing diseases in comparison with experienced specialists [63,64].

AI has been applied in endodontics for detecting pulpal diseases. Tumbelaka et al. [28] published details of an AI model for identifying pulpitis. This model was very efficient in precisely diagnosing reversible and irreversible pulpitis. However, the authors suggested using digital radiographs in order to achieve better validation. Zheng et al. [44] investigated a DL model designed for detecting deep caries and pulpitis, and the model demonstrated excellent performance. The ResNet18 model displayed outstanding performance when compared with reference models and experienced clinicians. However, this study focused only on teeth with single carious lesions and not on multiple carious lesions. Hence, further clinical validation is required before application in clinical practice.

Untreated dental caries progresses into periapical diseases, which are a result of the inflammatory lesions affecting the pulpal and periapical tissues, 90% of which are classified as apical granulomas, apical cysts, or abscesses [65]. The prevalence of apical periodontitis ranges between 34 and 61%, followed by periapical cysts and granulomas which range from 6 to 55% and from 46 to 94%, respectively [66,67,68]. Periapical pathosis can be detected radiographically as periapical radiolucencies, which are also termed apical lesions. Detecting apical lesions using radiographs is a daily task of clinicians; however, regardless of their discriminatory ability, radiographic examinations are influenced by inter- and intra-examiner reliability [69,70]. Ekert et al. [31] described the application of an AI model for detecting apical lesions; this model displayed satisfactory ability in detecting apical lesions, with an AUC of 0.85 and a sensitivity of 0.65. However, the sensitivity of the model was limited and needs to be improved by using a larger number of datasets to avoid the under-detection of the lesions before the model can be applied in clinics. Setzer et al. [35] described an AI model designed for segmenting CBCT images and detecting periapical lesions. The model displayed excellent accuracy and specificity. However, the limitation of this study was the comparison of the performance of the CNN model with clinicians’ segmentation, which can be subject to human error. Another limitation was the lower Dice index ratios for segmentation of the label lesions, which need to be addressed by increasing the training size. Orhan et al. [36] described an AI model designed for detecting periapical pathosis; the model displayed outstanding reliability in correctly detecting periapical lesions, which was equivalent to the performance of human experts. However, the presence of endo-perio lesions and periodontal defects can alter the performance of the model. In addition, anatomical structures such as the mental foramen and nasal fossa need segmentation, which can impact the analysis of the models’ measurements, and therefore, further programming will be required to address these issues. Endres et al. [37] reported the performance of an AI model designed for detecting periapical disease which displayed an acceptable precision and F1 score. The model achieved a better performance than experienced specialists. However, the model was trained using datasets labeled by the surgeons, which can be subject to human bias and be reflected in a degradation of the model performance. Another limitation was with the data used for training and evaluating the model, which were from a single center. Hence, further tests may be required with data from multiple centers to demonstrate generalizability. Li et al. [38]. studied the performance of a DL model designed for detecting apical lesions. The model demonstrated an excellent diagnostic accuracy of 92.5%. This model displayed a performance superior to that of a previous model [36]. However, the limitation of this model was with datasets that were obtained from a single hospital. Again, in order to demonstrate the generalizability of these results, further research is required with data from multiple sources [40].

Pauwels et al. [44] described the performance of a DL model designed for detecting periapical lesions. The results of this study were very promising, with a mean sensitivity of 0.87, specificity of 0.98, and ROC-AUC of 0.93. This model outperformed in comparison with experienced oral radiologists. This model further needs to be trained and validated on large samples/clinical radiographs before implementation in clinical scenarios, since this study used bovine ribs and simulated lesions. Ngoc et al. [49] detailed the performance of an AI model for diagnosing periapical lesions. This model displayed exceptional performance in comparison with endodontists’ diagnoses. However, this model was developed with a limited number of datasets using periapical radiographs.

Kirnbauer et al. [50] described the performance of an AI model for automatically detecting periapical lesions; the model displayed a sensitivity of 97.1% and a specificity of 88.0% for lesion detection. Bayrakdar et al. [52] reported on an AI model designed for segmenting apical lesions. This model was efficient in evaluating the periapical pathology and displayed a remarkable performance. However, there were a few limitations with the radiographic data used for this study, as they were obtained from a single piece of equipment and the number of samples used was very limited. Calazans et al. [55] reported on AI models for classifying periapical lesions and compared their performance with that of experienced oral and maxillofacial radiologists. The model displayed an accuracy of 70% and specificity of 92.39%, which were superior to those of the AI model VGG-16 and human experts.

Determining the working length is one of the crucial clinical steps that influence the outcome of root canal treatment. This will reduce the chances of insufficient cleaning of the canal and help in confining the root canal filling material into the canal and not invading the periapical tissues, ultimately resulting in a successful treatment outcome [70]. Saghiri et al. [25] described the performance of an AI-based model for locating the minor apical foramen. This model demonstrated good accuracy in detecting the apical foramen. Saghiri et al. [26] also described the performance of an AI model for determining the working length. The AI model demonstrated 96% accuracy in comparison with experienced endodontists. However, the quality of patient selection in these studies was low since the samples used were extracted teeth and cadavers. Qiao et al. [38] described the performance of an AI model designed for root canal length measurement. The accuracy of the model was exceptional and was better than the accuracy of the dual-frequency impedance ratio method, which demonstrated an accuracy of 85%. However, very limited samples were used, and increasing the sample size in future studies can further enhance the performance.

VRFs are crack types that can be complete or incomplete fractures of the root in the longitudinal plane and can be seen in teeth that are either endodontically treated or untreated [71,72]. These fractures are often unnoticed by clinicians and in most cases are only thought of when significant periapical changes occur, ultimately resulting in a delay in diagnosis and treatment [73]. To increase the diagnostic efficiency of clinicians, AI models have been applied for assisting clinicians in the early diagnosis of tooth cracks and fractures. Kositbowornchai et al. [27] described the performance of an AI model designed for detecting VRFs, and the model displayed an outstanding performance. However, the limitation of this study was with the samples, since they only used single-rooted premolar teeth; thus, these results cannot be generalized unless applied to different tooth types. Johari et al. [29] described the performance of an AI model for determining VRFs; the model displayed exceptional performance. However, in this study, only single-rooted premolar teeth were used. These findings were similar to the findings of the study conducted by Fukuda et al. [32] in which the AI model displayed a precision of 0.93 and an *F* measure of 0.83. However, the limitation of this study was with the datasets used, which were only from a single center, and only the radiographs with clear VRF lines were included [32]. Hu et al. [60] described the performance of AI models for diagnosing VRFs; the ResNet50 model presented the highest accuracy and sensitivity for diagnosing VRF teeth. Shah et al. [30] described the application of an AI model for automatically detecting cracks in teeth; this model displayed a mean ROC of 0.97 in detecting cracked teeth.

Assessing the shape of the roots and canals of a tooth can be very important in successfully treating a carious tooth. However, the variations in the root canal morphology pose a difficulty in canal preparation, irrigation, and obturation. C-shaped canals are the most difficult variation in the performance of a root canal treatment [74,75]. Hiraiwa et al. [32] described the application of an AI model designed for assessing the root morphology of the mandibular first molar; this model displayed an accuracy of 86.9%. Sherwood et al. [39] also described the performance of a DL model for classifying C-shaped canal anatomy in mandibular second molars. Both Xception U-Net and residual U-Net performed significantly better than the U-Net model. However, the limited sample used in this study and the focus on only C-shaped root canal anatomy were limitations of the study. Jeon et al. [45] reported on a DL model designed for predicting C-shaped canals in mandibular second molars; the model displayed outstanding performance in predicting C-shaped canals. Yang et al. [58] described the performance of a DL model for classifying C-shaped canals in mandibular second molars; the model displayed excellent performance in predicting C-shaped canals in both periapical and panoramic images. However, in this study, the number of samples used was insufficient, and the samples were from a single center.

AI has also been applied in predicting the prognosis of RCT. Herbst et al. [51] reported on an AI model for predicting factors associated with the failure of root canal treatments. This model was efficient in predicting tooth-level factors. Qu et al. [58] described the application of machine learning models for the prognosis prediction of endodontic microsurgery. The gradient boosting machine (GBM) model displayed excellent performance. These findings were similar to the finding of the study conducted by Li et al. [59] in which the model displayed an accuracy of 57.96–90.20%, an AUC of 95.63%, and a sensitivity of 91.39%. These automated models can be of great value to clinicians by assisting them in decision-making, providing quick and accurate results, overcoming the requirement of high-level clinical experience, and avoiding inter-observer variability.

The findings of this systematic review show that the majority of the AI models are designed for automated digital diagnosis and treatment planning. These findings are in accordance with the systematic reviews that have previously reported on various disciplines of dentistry. Mohammad-Rahimi et al. [76] reported on the performance of deep learning models in periodontology and oral implantology, where the authors concluded that the performance of the models is generally high. Albalawi et al. [77] reported on a wide range of AI models applied in orthodontics and concluded that these AI models are reliable and can automatically complete tasks with an enhanced speed and an efficiency equivalent to that of experienced specialists. Junaid et al. [78] reported on the application and performance of AI models designed for cephalometric landmark identification. The authors concluded that these models are of great benefit to orthodontists as they can perform tasks very efficiently. Carrillo-Perez et al. [79] reported on the application of AI models in dentistry; the authors concluded that the AI models display outstanding performance in performing the tasks. Thurzo et al. [80] reported on a wide range of AI models that have been designed for application in dentistry. The authors reported that there has been extraordinary growth in the development of AI models designed for application in dentistry. In the last few years, significant growth has been witnessed in the application of AI in dentistry.

This systematic review might have a few limitations. Even though we performed a comprehensive search for articles that have reported on the application of AI models in endodontics, we might have missed a few. Another limitation could be with the assessment of the risk of bias, which might vary between subjective judgments. However, considering the potential of AI applications in improving the diagnosis and treatment outcomes in endodontics, regulatory bodies should expedite the process of policy-making, approval, and marketing of these products for application in clinical scenarios.

## 5. Conclusions

In endodontics, AI models have been applied for determining working length, vertical root fractures, and root morphology; detecting and diagnosing pulpal diseases and periapical lesions; and predicting prognosis, postoperative pain, and case difficulties. Most of the included studies (*n* = 21) were developed using convolutional neural networks. Among the included studies, datasets that were used were mostly cone-beam computed tomography images, followed by periapical radiographs and panoramic radiographs. QUADAS-2, used to assess the quality of the included articles, revealed a low risk of bias in the patient selection domain in most of the studies (risk of bias: 90%; applicability: 70%). These models can be used as supplementary tools in clinical practice in order to expedite the clinical decision-making process and enhance the treatment modality and clinical operation. However, in most of the studies, the models were developed using a limited number of datasets for training and evaluation. The data samples collected were from a single clinic/center and from a single radiographic instrument. Hence, the results obtained from these studies cannot be generalized due to the lack of heterogeneity in the samples. In order to overcome these limitations, future studies should focus on considering a large number of datasets for training and testing the models. Samples need to be collected from multiple centers and from different radiographic instruments.

## Figures and Tables

**Figure 1 diagnostics-13-00414-f001:**
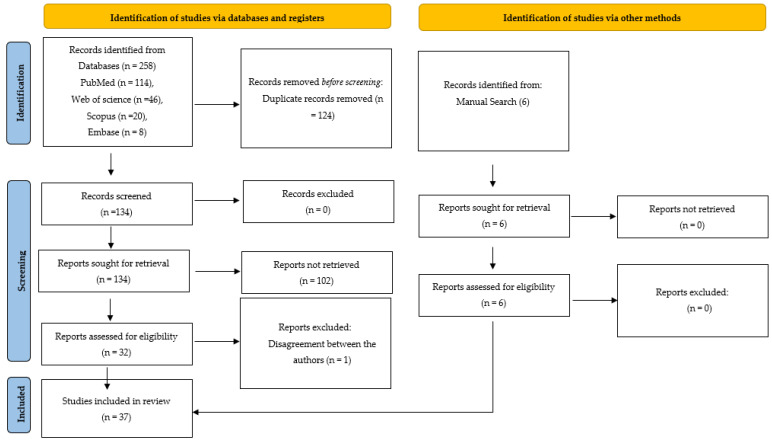
PRISMA 2020 flow diagram for new systematic reviews which included searches of databases, registers, and other sources.

**Figure 2 diagnostics-13-00414-f002:**
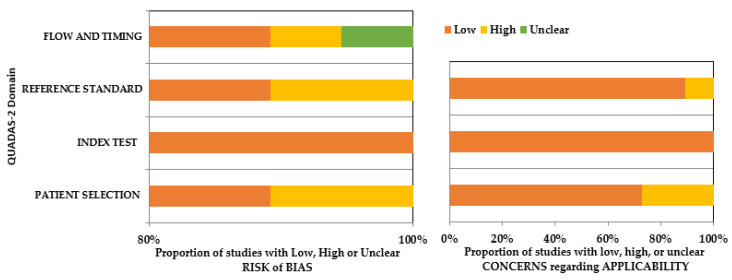
QUADAS-2 assessment of the individual risk of bias domains and applicability concerns.

**Table 1 diagnostics-13-00414-t001:** Structured search strategy carried out in electronic databases.

Search/Filters	Topic and Terms
“English” Language	“artificial intelligence” OR “automatic learning” OR “supervised learning” OR “unsupervised learning” OR “deep learning” OR “machine learning” OR “neural networks” OR “convolutional neural network” OR “computer assisted diagnosis” “endodontic dentistry” OR “root canal treatment” OR “apical lesions” OR “periapical lesions” OR “periapical pathology” OR “periapical diseases” OR “deep caries detection” OR “tooth segmentation” OR “pulp cavity segmentation” OR “root segmentation” OR “root morphology” OR “canal shape” OR “cracked tooth” OR “tooth fractures” OR “root fractures” OR “accuracy” OR “prediction” OR “diagnosis” OR ”expert systems” OR ” fuzzy networks” OR ” AI networks” OR “ AI models”
“English” Language	“artificial intelligence” AND “automatic learning” AND “supervised learning” AND “unsupervised learning” AND “deep learning” AND “machine learning” AND “neural networks” AND “convolutional neural network” AND “computer assisted diagnosis” “endodontic dentistry” AND “root canal treatment” AND “apical lesions” AND “periapical lesions” AND “periapical pathology” AND “periapical diseases” AND “deep caries detection” AND “tooth segmentation” AND “pulp cavity segmentation” AND “root segmentation” AND “root morphology” AND “canal shape” AND “cracked tooth” AND “tooth fractures” AND “root fractures” AND “accuracy” AND “prediction” AND “diagnosis” AND “expert systems” AND “fuzzy networks” AND “AI networks” AND “AI models”

**Table 2 diagnostics-13-00414-t002:** Details of the studies that have reported on the application of AI-based models in endodontics.

Serial No.	Authors	Year of Publication	Study Design	Algorithm Architecture	Objective of the Study	No. of Patients/Images/Photographs for Testing	Study Factor	Modality	Comparison If Any	Evaluation Accuracy/Average Accuracy/Statistical Significance	Results: (+) Effective, (−) Non-Effective, (N) Neutral	Outcomes	Authors’ Suggestions/Conclusions
1	Saghiri et al. [25]	2011	Comparative study	ANNs	AI-based model for locating the minor apical foramen	50 samples	Apical foramen	Intraoral radiographs	Two experienced endodontists	For 93% of the samples, the model determined the location of the apical foramen correctly	(+) Effective	ANN-based model demonstrated good accuracy in detecting the apical foramen	The AI model can be useful for secondary opinion in order to achieve better clinical decision-making
2	Saghiri et al. [26]	2012	Comparative study	ANNs	AI-based model for determining the working length	50 samples	Working length	Intraoral radiographs	Dentist	AI model demonstrated 96% accuracy in comparison with the experienced endodontists whose accuracy was 76%	(+) Effective	AI model demonstrated more accuracy in determining the working length in comparison with experienced endodontists	This model was efficient in determining the working length
3	Kositbowornchai et al. [27]	2013	Comparative study	ANNs	AI-based model for determining vertical root fracture (VRFs)	200 samples (80 for training, 120 for testing)	Vertical root fracture	Digital radiographs	Between groups	Sensitivity (98%), specificity (90.5%), and accuracy (95.7%)	(+) Effective	This AI model displayed sufficient sensitivity, specificity, and accuracy	This model make can be useful for making correct interpretations of root fractures
4	Tumbelaka et al. [28]	2014	Observational study	PNNs	AI-based model for identifying pulpitis	20 samples	Pulpitis	Periapical radiographs	None	Mean square error around 0.0003	(+) Effective	This model precisely diagnosed reversible and irreversible pulpitis	In order to obtain a better diagnosis, radiographs have to be digitalized
5	Johari et al. [29]	2017	Comparative study	Probabilistic neural networks (PNNs)	AI-based model for diagnosing VRFs in intact and endodontically treated teeth	240 samples	Vertical root fracture	CBCT images	Other state-of-the-art approaches	Accuracy of 96.6%, sensitivity of 93.3%, and specificity of 100%	(+) Effective	This model is efficient in diagnosing VRFs using CBCT images	Additional training of AI-based models is required before clinical use
6	Shah et al. [30]	2018	Comparative study	CNNs	AI model for automatically detecting cracks in teeth	6 samples	Cracked teeth	CBCT images	Frangi’s vessel enhancement algorithm	Mean ROC was 0.97	(+) Effective	This model was efficient in detecting cracked teeth	The model can detect cracked teeth in earlier stages and prevent pain and suffering associated with them
7	Ekert et al. [31]	2019	Comparative study	CNNs	AI model for detecting apical lesions	85 samples	Apical lesions	Panoramic radiographs	6 independent examiners	AUC was 0.85 (0.04). Sensitivity was 0.65 (0.12) and specificity was 0.87 (0.04)	(+) Effective	This model showed a satisfying ability to detect apical lesions	Sensitivity of the model needs to be improved before application in clinics
8	Fukuda et al. [32]	2019	Comparative study	CNNs	AI model for detecting vertical root fractures (VRFs)	300 samples (240 for training and 60 for testing)	Vertical root fracture	Panoramic radiographs	2 radiologists and 1 endodontist	Recall was 0.75, precision was 0.93, and *F* measure was 0.83	(+) Effective	This model showed promising results in detecting VRFs	This model has to be trained and applied on datasets from other hospitals
9	Hiraiwa et al. [33]	2019	Comparative study	ANNs	AI model for assessing the root morphology of the mandibular first molar	760 samples	Root morphology	Panoramic and CBCT images	Radiologist	Accuracy of 86.9%	(+) Effective	The model displayed high accuracy in diagnosing a single or extra root in the distal roots of mandibular first molars	This model displayed a high level of diagnostic ability
10	Mallishery et al. [34]	2019	Comparative study	ML	AI-based ML model for predicting the difficulty level of the case	500 samples	Case difficulty	Datasets	2 endodontists	Sensitivity of 94.96%	(+) Effective	This model displayed an excellent prediction of case difficulty	This model displayed excellent prediction ability which can increase the speed of decision-making and referrals
11	Setzer et al. [35]	2020	Comparative study	CNNs	A deep learning model for automated segmentation of CBCT images and detecting periapical lesions	20 CBCT images (16 CBCT images for training, 4 CBCT images for validation)	Apical lesions	CBCT images	1 radiologist, 1 endodontist, and 1 senior graduate	Accuracy of 0.93 and specificity of 0.88	(+) Effective	With a limited CBCT training, this model displayed excellent results in detecting the lesion	This model can aid clinicians with automated lesion detection
12	Orhan et al. [36]	2020	Comparative study	ANNs	AI model for detecting periapical pathosis	153 samples	Periapical lesions	CBCT images	1 maxillofacial radiologist	Reliability of 92.8% in correctly detecting periapical lesions	(+) Effective	There was no difference in the accuracy of humans and AI model in detecting apical lesions	This model will be useful for detecting periapical pathosis in clinical scenarios
13	Endres et al. [37]	2020	Comparative study	CNNs	Deep learning model for detecting periapical disease	197 samples (95 images for training and 102 images for testing)	Periapical disease	Panoramic radiographs	24 oral and maxillofacial (OMF) surgeons	Average precision of 0.60 and an F1 score of 0.58	(+) Effective	This deep learning algorithm achieved a better performance than 14 of 24 OMF surgeons	The deep learning model has potential to assist OMF surgeons in detecting periapical lucencies
14	Qiao et al. [38]	2021	Comparative study	CNNs	Deep learning models for root canal length measurement	21 samples	Root canal length	Tooth	Dual-frequency impedance ratio method	Accuracy of 95%	(+) Effective	This model demonstrated better accuracy in comparison with other models	The performance of this model can be enhanced by increasing the number of samples
15	Sherwood et al. [39]	2021	Comparative study	CNNs	Deep learning model for classifying C-shaped canal anatomy in mandibular second molars	135 samples (100 images for training and 35 images for testing)	Canal shapes	CBCT images	U-Net, residual U-Net, and Xception U-Net architectures	The mean Dice coefficients were 0.768 ± 0.0349 for Xception U-Net, 0.736 ± 0.0297 for residual U-Net, and 0.660 ± 0.0354 for U-Net on the test dataset	(+) Effective	Both Xception U-Net and residual U-Net performed significantly better than U-Net	Deep learning models can aid clinicians in detecting and classifying C-shaped canal anatomy
16	Li et al. [40]	2021	Comparative study	CNNs	Deep learning model for detecting apical lesions	460 samples (322 images for training and 138 images for testing)	Apical lesions	Periapical radiographs	3 experienced dentists	Diagnostic accuracy of the model was 92.5%	(+) Effective	Deep neural models demonstrated excellent accuracy in detecting the periapical lesions	This automated model allows dentists to make the diagnosis process shorter and more efficient
17	Vicory et al. [41]	2021	Comparative study	ML	AI model for detecting tooth microfractures	36 samples	Tooth microfractures	High-resolution (hr) CBCT and micro-CT scans	Direction–projection–permutation	Significant separation result	(+) Effective	The data suggest that this approach can be applied to hr-CBCT (clinically) when the images are not over-processed	Early detection of microfractures can help in planning appropriate treatment
18	Zheng et al. [42]	2021	Comparative study	CNNs	Deep learning model for detecting deep caries and pulpitis	844 samples (717 images for training and 127 images for testing)	Deep caries and pulpitis	Periapical radiographs	VGG19, Inception V3, ResNet18, 5 experienced dentists	Accuracy of 0.86, precision of 0.85, sensitivity of 0.86, specificity of 0.86, and AUC of 0.94	(+) Effective	ResNet18 demonstrated the best performance also in comparison with experienced dentists	The promising potential of this model can be applied for clinical diagnosis
19	Moidu et al. [42]	2021	Comparative study	CNNs	Deep learning model for categorization of endodontic lesions	1950 samples	Periapical lesions	Periapical radiographs	3 endodontists	Sensitivity/recall of 92.1%, 76% specificity, 86.4% positive predictive value, and 86.1% negative predictive value	(+) Effective	The model exhibited excellent sensitivity, positive predictive value, and negative predictive value	This AI model can be beneficial for clinicians and researchers
20	Pauwels et al. [44]	2021	Comparative study	CNNs	Deep learning model for detecting periapical lesions	10 samples	Periapical lesions	Periapical radiographs	3 oral radiologists	Mean sensitivity of 0.87, specificity of 0.98, and ROC-AUC of 0.93	(+) Effective	This CNN model displayed perfect accuracy for the validation data	This model showed promising results in detecting periapical lesions
21	Jeon et al. [45]	2021	Comparative study	CNNs	Deep learning model for predicting C-shaped canals in mandibular second molars	2040 samples (1632 images for training and 408 images for testing)	C-shaped canals	Panoramic radiographs	1 experienced radiologist and 1 experienced endodontist	Accuracy of 95.1, sensitivity of 92.7, specificity of 97.0, and precision of 95.9%	(+) Effective	This CNN model displayed significant accuracy in predicting C-shaped canals	This model can assist clinicians with dental image interpretation
22	Guo et al. [46]	2021	Comparative study	ANNs	Radial basis function neural network (RBFN)-based AI model for predicting thrust force and torque for root canal preparation	2 samples	Thrust force and torque	CT scans	Comparative ANN model	Prediction error less than 14%	(+) Effective	This model displayed an excellent prediction of thrust force and torque in canal preparation	Can be useful for instructing dentists during root canal preparations and also for improving the geometrical design of nickel titanium files
23	Lin et al. [47]	2021	Comparative study	ANNs	AI model for automatic and accurate segmentation of the pulp cavity and tooth	30 samples (25 sets for training and 5 sets for testing)	Segmentation of the pulp cavity	CBCT images	1 experienced endodontist	Dice similarity coefficient of 96.20% ± 0.58%, precision rate of 97.31% ± 0.38%, recall rate of 95.11% ± 0.97%, average symmetric surface distance of 0.09 ± 0.01 mm, and Hausdorff distance of 1.54 ± 0.51 mm in the tooth and Dice similarity coefficient of 86.75% ± 2.42%, precision rate of 84.45% ± 7.77%, recall rate of 89.94% ± 4.56%, average symmetric surface distance of 0.08 ± 0.02 mm, and Hausdorff distance 1.99 ± 0.67 mm in the pulp cavity	(+) Effective	The analysis performed by the model was better than that of the experienced endodontist	This model demonstrated excellent accuracy and hence can be applied in research and clinical tasks in order to achieve better endodontic diagnosis and therapy
24	Gao et al. [48]	2021	Observational study	ANNs	Backpropagation (BP) AI model for predicting postoperative pain following root canal treatment	300 samples (210 for training, 45 for validating, and 45 for testing)	Postoperative pain	Datasets	None	Accuracy of prediction was 95.60%	(+) Effective	This model displayed an excellent prediction of postoperative pain following RCT	The results displayed by this model have shown clinical feasibility and clinical application value
25	Ngoc et al. [49]	2021	Comparative study	CNNs	AI-based model for diagnosis of periapical lesions	130 samples	Periapical lesions	Bitewing images	Endodontists	Sensitivity of 89.5, specificity of 97.9, and accuracy of 95.6%	(+) Effective	This model displayed excellent performance and can be used as a support tool in the diagnosis of periapical lesions	This model can be used in teledentistry for the diagnosis of periapical diseases where there is a lack of dentists
26	Kirnbauer et al. [50]	2022	Observational study	CNNs	AI model for the automated detection of periapical lesions	144 samples	Periapical lesions	CBCT images	None	Sensitivity of 97.1% and specificity of 88.0% for lesion detection	(+) Effective	This AI model displayed excellent results compared with related literature	This model can be applied for testing under clinical conditions
27	Herbst et al. [51]	2022	Comparative study	ML	AI-based ML model for predicting failure of root canal treatment	591 samples	Root canal failure	Datasets	Random forest, gradient boosting machine, extreme gradient boosting, predictive modeling	logR 0.63, gradient boosting machine (GBM) 0.59, random forest (RF) 0.59, extreme gradient boosting (XGB) 0.60	(N) Neutral	This study found tooth-level factors to be associated with failure	With this AI model, predicting failure was only limitedly possible
28	Bayrakdar et al. [52]	2022	Observational study	CNNs	AI-based deep convolutional neural network (D-CNN) model for the segmentation of apical lesions	470 samples	Apical lesions	Panoramic radiographs	None	Sensitivity of 0.92, precision of 0.84, and F1-score of 0.88	(+) Effective	This AI model was efficient in evaluating periapical pathology	This AI model may facilitate clinicians in the assessment of periapical pathology
29	Zhao et al. [53]	2022	Comparative study	CNNs	AI model for evaluating the curative effect after treatment of dental pulp disease (DPD)	120 samples	Dental pulp disease	Radiographs and CBCT images	Control group with healthy teeth	Segmentation accuracy was 85.5%; diagnostic rate of X-ray was 43.7% and diagnostic rate of CBCT was 100%	(+) Effective	CBCT evaluation using an AI model can be an effective method for evaluating the curative effect of dental pulp disease treatment during and after the surgery	This model has a higher application prospect in the diagnosis and treatment of DPD
30	Hamdan et al. [54]	2022	Comparative study	CNNs	AI model for detecting apical radiolucencies	68 samples	Apical radiolucencies	Periapical radiographs	Eight experienced specialists	Alternative free-response receiver operating characteristic (AFROC) of 0.892, specificity of 0.931, and sensitivity of 0.733	(+) Effective	This model has the potential to improve the diagnostic efficacy of clinicians	This AI model enhances clinicians’ abilities to detect apical radiolucencies
31	Calazans et al. [55]	2022	Comparative study	CNNs	AI models for classifying periapical lesions	1000 samples (training 60%, validation 20%, testing 20%)	Periapical lesions	CBCT scans	Experienced oral and maxillofacial radiologist	Accuracy of 70%, specificity of 92.39%	(+) Effective	DenseNet-121 network was superior to VGG-16 and human experts	The proposed models displayed a satisfactory classification performance
32	Yang et al. [56]	2022	Comparative study	CNNs	AI-based deep learning model for classifying C-shaped canals in mandibular second molars	1000 samples	C-shaped canals	Periapical and panoramic radiographs	Specialist and general clinician	AUC of 0.98 on periapical and AUC of 0.95 on panoramic	(+) Effective	This model displayed high accuracy in predicting the C-shaped canal in both periapical and panoramic images and was similar to the performance of a specialist and better than a general dentist	This model was effective in diagnosing C-shaped canals and therefore can be a valuable aid for clinicians and also in dental education
33	Xu et al. [57]	2022	Comparative study	ML	AI-based models for identifying the history of root canal therapy	920 samples (736 for training and 184 for testing)	Root canal therapy	Datasets	VGG16, VGG19, and ResNet50	Accuracies were above 95% and AUC area was 0.99	(+) Effective	This model displayed excellent accuracy and can aid in clinical auxiliary diagnosis based on image display	This AI-assisted diagnosis of oral medical images can be effectively promoted for clinical practice
34	Qu et al. [58]	2022	Comparative study	ML	AI-based machine learning models for predicting prognosis of endodontic microsurgery	234 samples (80% for the training set and 20% for the test set)	Predicting prognosis	Datasets	Gradient boosting machine (GBM) and random forest (RF) models	Accuracy of 0.80, sensitivity of 0.92, specificity of 0.71, positive predictive value (PPV) of 0.71, negative predictive value (NPV) of 0.92, F1 score of 0.80, and area under the curve (AUC) of 0.88	(+) Effective	The GBM model outperformed the RF model slightly on the dataset	The models can improve efficiency and assist clinicians in decision-making
35	Li et al. [59]	2022	Comparative study	ANNs	AI-based anatomy-guided multibranch transformer (AGMB-Transformer) network for assessing the result of root canal therapy	245 samples	Root canal therapy evaluation	Datasets	2 experienced specialists and other models (ResNet50, ResNeXt50, GCNet50, BoTNet50)	Accuracy ranged from 57.96% to 90.20%, AUC of 95.63%, sensitivity of 91.39%, specificity of 95.09%, F1 score of 90.48%	(+) Effective	This model achieved a highly accurate evaluation	The performance of this model has important clinical value in reducing the workload of endodontists
36	Hu et al. [60]	2022	Comparative study	CNNs	AI-based deep learning models for diagnosing vertical root fracture	276 samples	Vertical root fracture	CBCT images	2 experienced radiologists, ResNet50, VGG19, and DenseNet169	The accuracy, sensitivity, specificity, and AUC were 97.8%, 97.0%, 98.5%, and 0.99	(+) Effective	ResNet50 presented the highest accuracy and sensitivity for diagnosing VRF teeth	ResNet50 presented the highest diagnostic efficiency in comparison with other models. Hence, this model can be used as an auxiliary diagnostic technique to screen for VRF teeth
37	Vasdev et al. [61]	2022	Comparative study	CNNs	AI-based deep learning model for detecting healthy and non-healthy periapical images	16,000 samples	Periapical lesions	Periapical radiographs	ResNet-18, ResNet-34, and AlexNet	Accuracy of 0.852, precision and F1 score of 0.850	(+) Effective	This AlexNet model outperformed the other models	This model generalizes effectively to previously unseen data and can aid clinicians in diagnosing a variety of dental diseases

Footnotes: ML = machine learning, ANNs = artificial neural networks, CNNs = convolutional neural networks, DCNNs = deep neural networks, c-index = concordance index, CT = computed tomography, CBCT = cone-beam computed tomography, OCT = optical coherence tomography.

**Table 3 diagnostics-13-00414-t003:** Assessment of Strength of Evidence.

Outcome	Inconsistency	Indirectness	Imprecision	Risk of Bias	Publication Bias	Strength of Evidence
Application of AI for determining working length [25,26,38]	Not Present	Not Present	Not Present	Present	Not Present	⨁⨁⨁◯
Application of AI for determining vertical root fracture [27,29,30,32,41,60]	Not Present	Not Present	Not Present	Present	Not Present	⨁⨁⨁◯
Application of AI for detecting pulpal diseases [28,42]	Not Present	Not Present	Not Present	Present	Not Present	⨁⨁⨁◯
Application of AI for, detecting and diagnosing periapical lesions [31,35,36,37,40,43,44,49,50,52,54,55,61]	Not Present	Not Present	Not Present	Present	Not Present	⨁⨁⨁◯
Application of AI for determining root morphology [33,39,45,56]	Not Present	Not Present	Not Present	Not Present	Not Present	⨁⨁⨁⨁
Application of AI for determining root canal failures [51,57,58]	Not Present	Not Present	Not Present	Not Present	Not Present	⨁⨁⨁⨁
Application of AI for predicting postoperative pain [42]	Not Present	Not Present	Not Present	Present	Not Present	⨁⨁⨁◯
Application of AI for predicting case difficulty [34]	Not Present	Not Present	Not Present	Not present	Not Present	⨁⨁⨁⨁
Application of AI for determining thrust force and torque in canal preparation [46]	Not Present	Not Present	Not Present	Present	Not Present	⨁⨁⨁◯
Application of AI for segmenting pulp cavities [47]	Not Present	Not Present	Not Present	Not Present	Not Present	⨁⨁⨁⨁
Application of AI in curative effect after treatment [53,59]	Not Present	Not Present	Not Present	Not Present	Not Present	⨁⨁⨁⨁

⨁⨁⨁⨁ = high evidence; ⨁⨁⨁◯ = moderate evidence.

## Data Availability

Not applicable.

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
