# Peer review of "Developments and Performance of Artificial Intelligence Models Designed for Application in Endodontics: A Systematic Review"

_diagnostics, 2023, doi:10.3390/diagnostics13030414_

Round 1

Reviewer 1 Report

This is an interesting study based on a rigourous Introduction:

Lines 63-71: Redundant

2.1 Search strategy: The authors need to define C: Comparison.

3.4 Risk of bias assessment and applicability concerns: The authors considered the "Reference standard" as a criterion to evaluate the risk of bias, whereas it should be a factor that determines the inclusion or exclusion of the study. This would be the "comparison" in PICO.

References: There are some recent studies that the authors have not included in their review for example: 

Albitar L, Zhao T, Huang C, Mahdian M. Artificial Intelligence (AI) for Detection and Localization of Unobturated Second Mesial Buccal (MB2) Canals in Cone-Beam Computed Tomography (CBCT). Diagnostics (Basel). 2022 Dec 18;12(12):3214. doi: 10.3390/diagnostics12123214. PMID: 36553221; PMCID: PMC9777585.

Author Response

Dear Reviewer,
Greetings!

Firstly I would like to thank you for your valuable inputs, and would like to inform you that we have considered all the valuable comments suggested by you and have modified the manuscript as per your suggestions.

We have modified the manuscript to best of our knowledge, kindly consider the same and oblige,

Thank you and regards

Sincerely yours,

Dr. Sanjeev B Khanagar (Corresponding Author)

Reviewer 1

Comment: Lines 63-71: Redundant

Response: We appreciate your concern however, we feel this paragraph is important as it deals with disease prevalence and importance of diagnosis. So we kindly request you to look into this, aspect and consider it.

Comment: 2.1 Search strategy: The authors need to define C: Comparison.

Response: Thank you for this very important highlight, we have considered your valuable suggestion and provided the details in the manuscript.

Comment: 3.4 Risk of bias assessment and applicability concerns: The authors considered the "Reference standard" as a criterion to evaluate the risk of bias, whereas it should be a factor that determines the inclusion or exclusion of the study. This would be the "comparison" in PICO.

Response: We appreciate your concern however, Risk of bias was considered based on sampling, index test used for comparison, validated reference and uniformity of data collection (time and reference standard). [Whiting, P.F.; Rutjes, A.W.S.; Westwood, M.E.; Mallett, S.; Deeks, J.J.; Reitsma, J.B.; Leeflang, M.M.G.; Sterne, J.A.C.; Bossuyt, P.M.M.; QUADAS-2 Group. QUADAS-2: A Revised Tool for the Quality Assessment of Diagnostic Accuracy Studies. Ann. Intern. Med. 2011, 155, 529–536, doi:10.7326/0003-4819-155-8-201110180-00009]

Comment: References: There are some recent studies that the authors have not included in their review for example: 

Albitar L, Zhao T, Huang C, Mahdian M. Artificial Intelligence (AI) for Detection and Localization of Unobturated Second Mesial Buccal (MB2) Canals in Cone-Beam Computed Tomography (CBCT). Diagnostics (Basel). 2022 Dec 18;12(12):3214. doi: 10.3390/diagnostics12123214. PMID: 36553221; PMCID: PMC9777585.

Response: Thank you for your valuable suggestion however, data search process was conducted for articles published from January 1, 2000, until November 30, 2022. So kindly consider

Reviewer 2 Report

This systematic review aims to evaluate the performance of artificial intelligence (AI) models developed for use in endodontics. The authors searched several online databases for relevant articles published between 2000 and 2022 and included 37 original research articles in their review. The quality of the included studies was assessed using the QUADAS-2 guidelines, and the GRADE approach was used to assess the certainty of the evidence. The authors report that AI models have been developed for a number of clinical tasks in endodontics, including working length determination, pulp disease detection, periapical lesion detection and diagnosis, postoperative pain prediction, and pulp cavity segmentation. These models have proven efficient in performing these tasks and have the potential to improve clinical outcomes and expedite the decision-making process for clinicians.

Of the positive aspects of this systematic review. The authors conducted an extensive search of multiple databases and included a wide range of AI models in their review. They also used established tools to assess the quality and safety of the evidence.

However, there are also some limitations to this review. First, the authors didn't provide clear and complete inclusion and exclusion criteria for the studies they included in their review. It's not clear how they selected the 37 studies from the potentially hundreds of articles they found in their search. As they're now listed in lines 142-150, this is insufficient and needs further elaboration. A systematic review protocol describing the rationale, hypothesis, and planned methods of review should be part of the appendix to this paper.

In addition, the authors did not provide a clear overview of the characteristics of the included studies, such as the sample size, the type of AI model used, and the type of endodontic outcomes measured. This makes it difficult to interpret the results of the study. The authors could also have more adequately described the methodology of their systematic review. For example, they could explain in more detail how they selected the identified articles for inclusion in the study, how they extracted the data from the included studies, or how they summarized the results. This makes it difficult to assess the quality of the study. The authors could have made a clearer conclusion or recommendation based on the results of their review. While they note that AI models have the potential to improve clinical outcomes in endodontics, it is not clear how they reached this conclusion or what the implications are for clinical practice.To improve the quality of this review, the authors could elaborate on their inclusion and exclusion criteria and provide a contextual overview of the characteristics of the included studies. They should also provide more information on the methodology of their systematic review, including how they screened and extracted data from the included studies.

The chapter Discussion shall discuss more and focus more on the results of the review and shall compare the findings to other systematic reviews like for example https://doi.org/10.3390/healthcare10071269 . In the first paragraph of the Discussion on lines 318-321 where mentioning advanced  CNN based AI models consider referencing other recent papers on Use of Advanced Artificial Intelligence.

Finally, the authors should provide a clear conclusion and recommendation based on the results of their study - the current Conclusions chapter contains too much general indirect information. It should be more compact and summarize the results of this research.

The title needs to be corrected, use ":" instead of "-" and don't use the abbreviation "(AI)" in the title. The abstract should be rewritten to adequately reflect the results of this systematic review

Keywords must also be corrected. They mustn't contain words that are already mentioned in the title.

The record of this systematic review must be registered and available online in the Registry of Systematic Reviews/Meta-Analyses, or in the International Prospective Register of Systematic Reviews—PROSPERO.

After these major improvmentes the paper might be useful for clinical practise and can be published.

Author Response

Dear Reviewer,
Greetings of the day!

Firstly I would like to thank you for your valuable inputs, and would like to inform you that we have considered your valuable comments and have modified the manuscript as per your suggestions.

We are also providing point to point clarifications for the comments suggested by you. "Please see the attachment."

We have modified the manuscript to best of our knowledge, kindly consider the same and oblige,

Thank you and regards

Sincerely yours,

Dr. Sanjeev B Khanagar (Corresponding Author)

Reviewer 2

Comment: There are also some limitations to this review. First, the authors didn't provide clear and complete inclusion and exclusion criteria for the studies they included in their review. It's not clear how they selected the 37 studies from the potentially hundreds of articles they found in their search. As they're now listed in lines 142-150, this is insufficient and needs further elaboration.

Response: We have considered your valuable suggestions, and provided more details in the manuscript in the material and methods section.

Comment: A systematic review protocol describing the rationale, hypothesis, and planned methods of review should be part of the appendix to this paper.

Response: We have considered your valuable suggestions, and provided an approved protocol with complete details in the article submission process as an annexure along with the manuscript. Please find the screenshot of the same below.

Comment: In addition, the authors did not provide a clear overview of the characteristics of the included studies, such as the sample size, the type of AI model used, and the type of endodontic outcomes measured. This makes it difficult to interpret the results of the study.

Response: We appreciate your concern, we have provided all the details of the study characteristics in Table 2

Comment: The authors could also have more adequately described the methodology of their systematic review. For example, they could explain in more detail how they selected the identified articles for inclusion in the study, how they extracted the data from the included studies, or how they summarized the results. This makes it difficult to assess the quality of the study.

Response: We have considered your valuable suggestions, and provided more details in the manuscript in the material and methods section.

Comment: The authors could have made a clearer conclusion or recommendation based on the results of their review. While they note that AI models have the potential to improve clinical outcomes in endodontics, it is not clear how they reached this conclusion or what the implications are for clinical practice.

Response: We have considered your valuable suggestions, and revised the conclusions as per the suggestions.

Comment: To improve the quality of this review, the authors could elaborate on their inclusion and exclusion criteria and provide a contextual overview of the characteristics of the included studies. They should also provide more information on the methodology of their systematic review, including how they screened and extracted data from the included studies.

Response: We have considered your valuable suggestions, and provided more details in the manuscript in the material and methods section.

Comment: The chapter Discussion shall discuss more and focus more on the results of the review and shall compare the findings to other systematic reviews like for example https://doi.org/10.3390/healthcare10071269. In the first paragraph of the Discussion on lines 318-321 where mentioning advanced CNN based AI models consider referencing other recent papers on Use of Advanced Artificial Intelligence.

Response: Thank you for your valuable suggestion and providing us a very important reference article. We have considered your valuable suggestions, and revised the discussion section.

Comment: Finally, the authors should provide a clear conclusion and recommendation based on the results of their study - the current Conclusions chapter contains too much general indirect information. It should be more compact and summarize the results of this research.

Response: Thank you for your valuable suggestion, we have revised the conclusion and recommendations based on our findings.

Comment: The title needs to be corrected, use ":" instead of "-" and don't use the abbreviation "(AI)" in the title. The abstract should be rewritten to adequately reflect the results of this systematic review

Response: We have considered your valuable suggestions, and made the changes in the title and abstract.

Comment: Keywords must also be corrected. They mustn't contain words that are already mentioned in the title.

Response: We have considered your valuable suggestions, and deleted the keywords that were already mentioned in the title.

Comment: The record of this systematic review must be registered and available online in the Registry of Systematic Reviews/Meta-Analyses, or in the International Prospective Register of Systematic Reviews—PROSPERO.

Response: We had applied for the PROSPERO registration, but unfortunately we were not successful

Reviewer 3 Report

Dear authors,

I assessed the article “Developments and performance of artificial Intelligence (AI) models designed for application in Endodontics - a Systematic Review”, which aimed to report the application and performances of AI models that have been designed for application in endodontics.

The theme is very interesting. Therefore, many concerns were raised.

ABSTRACT

- poorly written. Redo it.

INTRO: it is poorly presented; must be improved

- all the state of the art and background for AI were exposed between lines 86-90: “Additionally, in dentistry AI models are designed for diagnosing oral diseases like dental caries, periodontal diseases, and oral cancer as well as treatment planning for orthognathic surgeries and predicting the treatment outcomes [23-25]. These models have demonstrated excellent performances, with a major advantage of this being improved diagnostic efficiency with a reduced image interpretation time [26].”

I considered it completely poor. There is no reference (INTRO) that was applied for AI in endo. This fact led me to do questions about the feasibility of the study.

- unclude information about the types/modalities of AI currently exist

M&M:

- there was approval for the systematic review. Lines 95,96: “[Institutional Review Board Approval No-2439-22] before initiating the literature search process for this systematic review”

It is wrong.

RESULTS and DISCUSSION are well-done.

- correct the authors citation (last name et al., year)

CONCLUSION

- long text (reduce it)

- there is a mistake (review the conclusion)

Author Response

Dear Reviewer,
Greetings of the day!

Firstly I would like to thank you for your valuable inputs, and would like to inform you that we have considered your valuable comments and have modified the manuscript as per your suggestions.

We are also providing point to point clarifications for the comments suggested by you. "Please see the attachment."

We have modified the manuscript to best of our knowledge, kindly consider the same and oblige,

Thank you and regards

Sincerely yours,

Dr. Sanjeev B Khanagar (Corresponding Author)

Reviewer 3

Comment: ABSTRACT- poorly written. Redo it.

Response: We have considered your valuable suggestions, and revised the abstract.

Comment: INTRO: it is poorly presented; must be improved - all the state of the art and background for AI were exposed between lines 86-90: “Additionally, in dentistry AI models are designed for diagnosing oral diseases like dental caries, periodontal diseases, and oral cancer as well as treatment planning for orthognathic surgeries and predicting the treatment outcomes [23-25]. These models have demonstrated excellent performances, with a major advantage of this being improved diagnostic efficiency with a reduced image interpretation time [26].”

I considered it completely poor. There is no reference (INTRO) that was applied for AI in endo. This fact led me to do questions about the feasibility of the study.

- unclude information about the types/modalities of AI currently exist

Response: We appreciate your concern, however we have mentioned all the details of the articles that have reported on the application of AI models in endodontics in the results and discussion section. We have provided the details of types and modalities in AI in introduction section [lines 73-92]

Comment: M&M:- there was approval for the systematic review. Lines 95,96: “[Institutional Review Board Approval No-2439-22] before initiating the literature search process for this systematic review”

It is wrong.

Response: We appreciate your concern, we have mentioned it as it is a mandate to obtain an ethical clearance from our institutional review board.  Please find the screenshot of the same below.

Comment: RESULTS and DISCUSSION are well-done.- correct the authors citation (last name et al., year)

Response: Thank you for your kind words, we have considered your valuable suggestions, and revised the author citation.

Comment: CONCLUSION - long text (reduce it) - there is a mistake (review the conclusion)

Response: Thank you for your valuable suggestion, we have revised the conclusion section

Round 2

Reviewer 1 Report

I appreciate the authors' efforts to address the comments. However, I still believe that the third paragraph in the introduction provides information that has already been alluded to in the previous two paragraphs. If the authors wish to keep it, I recommend considering revising it.

The conclusion is too long. The first few sentences are re-iteration of facts that are best suited in the background section. The conclusion should only focus on facts derived from the results of the present study.

Author Response

Dear Reviewer,
Greetings!

Firstly I would like to thank you for your valuable inputs, and would like to inform you that we have considered all the valuable comments suggested by you and have modified the manuscript as per your suggestions.

We have modified the manuscript to best of our knowledge, kindly consider the same and oblige,

Thank you and regards

Sincerely yours,

Dr. Sanjeev B Khanagar (Corresponding Author)

Reviewer 1

Comment: I appreciate the authors' efforts to address the comments. However, I still believe that the third paragraph in the introduction provides information that has already been alluded to in the previous two paragraphs. If the authors wish to keep it, I recommend considering revising it.

Response: Thank you for your kind words of appreciation, we have considered your valuable suggestion and deleted the paragraph.

Comment: The conclusion is too long. The first few sentences are re-iteration of facts that are best suited in the background section. The conclusion should only focus on facts derived from the results of the present study.

Response: We have considered your valuable suggestion and revised the conclusion as per your suggestion

Reviewer 2 Report

Authors have improved the manuscript sufficiently.

Author Response

Dear Reviewer,
Greetings of the day!

Firstly I would like to thank you for your valuable inputs, and would like to inform you that we have considered your valuable comments and have modified the manuscript as per your suggestions.

We are also providing point to point clarifications for the comments suggested by you. "Please see the attachment."

We have modified the manuscript to best of our knowledge, kindly consider the same and oblige,

Thank you and regards

Sincerely yours,

Dr. Sanjeev B Khanagar (Corresponding Author)

Reviewer 2

Comment: Authors have improved the manuscript sufficiently.

Response: We sincerely thank you for considering our revisions.

Reviewer 3 Report

Dear authors,
I assessed (2nd round) the article “Developments and performance of artificial intelligence (AI) models designed for application in Endodontics - a Systematic Review”, which aimed to report the application and performances of AI models that have been designed for application in endodontics.

I continue appreciating the study. It is very interesting.

Therefore, concerns were still raised (minor review).

1. The abstract was modified, but it was not enough yet. Introduce more details about the included studies.

2. I sent a question (below), and after I read the response, I raised the following question: If this paragraph was enough to do the intro about AI, then there is no necessity to do a systematic study. Otherwise, the authors included 37 articles about AI applied to Endo. Please, improve state of the art for this point in the Introduction.

My question: “INTRO is poorly presented; must be improved - all the state of the art and background for AI were exposed between lines 86-90: “Additionally, in dentistry AI models are designed for diagnosing oral diseases like dental caries, periodontal diseases, and oral cancer as well as treatment planning for orthognathic surgeries and predicting the treatment outcomes [23-25]. These models have demonstrated excellent performances, with a major advantage of this being improved diagnostic efficiency with a reduced image interpretation time [26].”

I considered it completely poor. There is no reference (INTRO) that was applied for AI in endo. This fact led me to do questions about the feasibility of the study.
- include information about the types/modalities of AI currently exist”
Authors’ response: “We appreciate your concern, however we have mentioned all the details of the articles that have reported on the application of AI models in endodontics in the results and discussion section. We have provided the details of types and modalities in AI in introduction section [lines 73-92]”

3. IRB approval. It is the first time I have seen an ethical committee approve a systematic study. I saw and read the document submitted, and I continued asking myself about what would be the ethical concern. Therefore, it is not relevant; just a comment.

Author Response

Dear Reviewer,
Greetings of the day!

Firstly I would like to thank you for your valuable inputs, and would like to inform you that we have considered your valuable comments and have modified the manuscript as per your suggestions.

We are also providing point to point clarifications for the comments suggested by you. "Please see the attachment."

We have modified the manuscript to best of our knowledge, kindly consider the same and oblige,

Thank you and regards

Sincerely yours,

Dr. Sanjeev B Khanagar (Corresponding Author)

Reviewer 3

Comment: The abstract was modified, but it was not enough yet. Introduce more details about the included studies.
Response: Thank you for your valuable suggestion, we have revised the abstract as per your suggestion.

Comment: I sent a question (below), and after I read the response, I raised the following question: If this paragraph was enough to do the intro about AI, then there is no necessity to do a systematic study. Otherwise, the authors included 37 articles about AI applied to Endo. Please, improve state of the art for this point in the Introduction.

Response: We thank you for your valuable suggestion, we have added the details as per your suggestion

Comment: IRB approval. It is the first time I have seen an ethical committee approve a systematic study. I saw and read the document submitted, and I continued asking myself about what would be the ethical concern. Therefore, it is not relevant; just a comment.

Response: We appreciate your concern, but obtaining an ethical clearance from our institutional review board is mandatory in our institution.  Kindly consider.
